# Evaluation of the Cytotoxic Effect of Pd_2_Spm against Prostate Cancer through Vibrational Microspectroscopies

**DOI:** 10.3390/ijms24031888

**Published:** 2023-01-18

**Authors:** Raquel C. Laginha, Clara B. Martins, Ana L. C. Brandão, Joana Marques, M. Paula M. Marques, Luís A. E. Batista de Carvalho, Inês P. Santos, Ana L. M. Batista de Carvalho

**Affiliations:** 1Molecular Physical-Chemistry R&D Unit, Department of Chemistry, University of Coimbra, 3004-535 Coimbra, Portugal; 2Department of Life Sciences, Faculty of Science and Technology, University of Coimbra, 3000-456 Coimbra, Portugal

**Keywords:** prostate cancer, cisplatin, palladium(II), Raman microspectroscopy, FTIR microspectroscopy

## Abstract

Regarding the development of new antineoplastic agents, with a view to assess the selective antitumoral potential which aims at causing irreversible damage to cancer cells while preserving the integrity of their healthy counterparts, it is essential to evaluate the cytotoxic effects in both healthy and malignant human cell lines. In this study, a complex with two Pd(II) centers linked by the biogenic polyamine spermine (Pd_2_Spm) was tested on healthy (PNT-2) and cancer (LNCaP and PC-3) prostate human cell lines, using cisplatin as a reference. To understand the mechanisms of action of both cisplatin and Pd_2_Spm at a molecular level, Fourier Transform Infrared (FTIR) and Raman microspectroscopies were used. Principal component analysis was applied to the vibrational data, revealing the major metabolic changes caused by each drug, which were found to rely on DNA, lipids, and proteins, acting as biomarkers of drug impact. The main changes were observed between the B-DNA native conformation and either Z-DNA or A-DNA, with a higher effect on lipids having been detected in the presence of cisplatin as compared to Pd_2_Spm. In turn, the Pd-agent showed a more significant impact on proteins.

## 1. Introduction

It is estimated that prostate cancer was responsible for more than 375,000 deaths in 2020. This is the second most common type of cancer, after lung cancer, the fifth with the highest mortality rate in individuals with prostate, and the first in prevalence in the last five years [1]. Prostate cancer can be classified under several category types: (1) adenocarcinoma, which may be divided into acinar or ductal adenocarcinomas; (2) transitional cell and (3) squamous cell carcinomas; (4) small cell prostate cancer; (5) sarcoma; and (6) lymphoma tumors, with acinar adenocarcinoma being the most common [2]. Prostate cancer screening of asymptomatic individuals with prostate is carried out through blood analysis to assess prostate specific antigen (PSA) levels followed or not by digital rectal examination. Patients with positive findings undergo transrectal imaging techniques (ultrasound or magnetic resonance) and biopsy. The histopathological analysis of the biopsy enables the conclusive diagnosis of the type of tumor [3].

The treatment of choice for prostate cancer patients depends on various factors, such as the type of tumor and its staging. When detected at an early stage, the patient can either be kept under active surveillance/watchful waiting or proceed to radiotherapy, which may be combined with adjuvant or neoadjuvant androgen suppression hormone therapy. This consists of reducing the levels of testosterone and dihydrotestosterone (DHT) as a way of inhibiting tumor growth. When there is no tumor growth reduction after hormonal castration (castration-resistant prostate cancer, CRPC), chemotherapy might be necessary. If treatment fails, patients may undergo prostatectomy [3,4].

There are several antineoplastic agents currently approved for clinical use against CRPC, such as docetaxel, which is frequently used in combination with prednisone or even in a cocktail with other drugs like darolutamide [5,6]. Prednisone may also be administered in combination with the well-known compound cabazitaxel to treat metastatic castration resistant prostate cancer (mCRPC) when patients have already undergone chemotherapy that included docetaxel and became resistant [7,8]. These types of chemotherapeutic drugs, docetaxel and cabazitaxel, belong to the category of microtubule targeting compounds, specifically taxane anticancer agents, with their mechanism of action relying on the microtubules’ stabilization or increased polymerization [9]. Additionally, there are studies with promising results for inorganic agents, namely with platinum(II) and palladium(II) metal centers, against metastatic and non-metastatic CRPC [4,10]. Platinum agents have a different mode of action when compared to antimitotic targeting agents. They form adducts with DNA by direct binding between the metal ion and the nitrogen from the purine and pyrimidine bases, leading to disruption of the replication mechanisms and inducing cell death by apoptosis. Cisplatin (*cis*-Pt(NH_3_)_2_Cl_2_) was first synthesized in 1845 by Michele Peyrone, its cell growth-inhibiting effect having been discovered serendipitously by Barnett Rosenberg in 1960. In the 1970s, it was tested and approved for clinical use to fight ovarian and metastatic testicular cancers. Due to the positive outcomes of cisplatin as a chemotherapeutic drug, the development of inorganic agents has been more significant over the years [11,12]. Nevertheless, drug toxicity, adverse side effects, and cellular resistance mechanisms are three of the faced challenges when using conventional metal-based anticancer drugs [3,4]. To suppress these limitations, in the last decades, several Pt-based compounds, such as Picoplatin (Pt(II), BTP-114 (cisplatin pro-drug) and Satraplatin (Pt(IV)—oral platinum analogue) have undergone clinical trials for prostate cancer treatment [13], although none of these have yet been approved for clinical use due to low efficacy or severe side-effects.

In order to overcome these deleterious side effects and improve efficacy, polynuclear Pt(II) and Pd(II) chelates with biogenic polyamines have been developed over the past years [14]. Linear alkyl polyamines, such as putrescine, spermidine, and spermine, are polycations essential for cell growth but are also recognized for their association to neoplastic processes. They promote higher flexibility to the metal complexes (relative to cisplatin or carboplatin) and higher rates of interstrand cross-links [15,16,17]. Besides the structural similarity of these transition metals, Pd(II) complexes are more reactive than Pt(II), which correlates to faster reactions and higher lability. This is corroborated by the ligand field theory, the higher energy splitting in 5d orbitals in Pt compared to 4d orbitals in Pd leads to less nucleus–electron interaction in d orbitals, enabling the donation of two electrons to the ligands, giving rise to more stable bonds, usually less labile [14,17,18,19].

For an in-depth knowledge of the complexes and determination of structural–activity relationships (SAR’s), vibrational spectroscopy techniques may be used. Fourier transform infrared (FTIR) and Raman are non-destructive, non-invasive, cost-effective, reproducible, and highly accurate tools, very suitable for understanding the impact of the chemotherapeutic agents on cells, at a molecular level. These two vibrational methods allow to understand a drug’s metabolic impact under different conditions (e.g., drug-treated, or untreated cells). Therefore, both FTIR and Raman have been extensively applied in biomedical research, particularly to analyze the biochemical composition of cell lines or tissues and to classify malignant and healthy tissue in the early-cancer diagnosis or surgical guidance, allowing a better understanding of the disease onset at the biological and chemical levels [20,21,22,23,24,25,26,27,28].

Within the various available healthy and carcinogenic prostate cell lines, three can be highlighted: (1) PNT-2 cell line constituted by normal prostatic-epithelial cells, thus considered a healthy cell line; (2) LNCaP cell line, which corresponds to prostate adenocarcinoma cells sensitive to androgen levels; and (3) PC-3 cell line, are CRPC epithelial cells of adenocarcinoma unresponsive to decreasing androgen levels [4,29]. Some studies have reported higher values of HIF1α protein in PC-3 than in LNCaP, since this is likely a CRPC contributively of metastatic and chemo-resistance, a protein which is responsible to regulate the transcription of DNA [30].

Vibrational spectroscopy has been applied to prostate cancer to distinguish normal from cancer tissues [31,32,33,34,35], allowing the identification of biochemical fingerprints especially related to DNA and lipids, leading to an accurate evaluation of the sample. For prostate cells lines, the high wavenumbers have been used to fully characterize the PC-3 cell line [36] and to study the use of lipid altering drugs against PC-3 and PNT-2 cell lines [37]. FTIR microspectroscopy was used to analyze and classify several prostate cancer cell lines [38], and a few microFTIR studies are to be found in the literature reporting the use of anticancer agents such as paclitaxel, doxorubicin, or methrotrexate against the PC-3 cell line [39,40].

This work aims at evaluating cell viability in PNT-2, LNCaP, and PC-3 prostate cell lines in the presence of cisplatin and Pd_2_Spm, a complex with two Pd(II) centers linked by the biogenic polyamine spermine (Figure 1), which has shown increased cytotoxic activity against several human cancer cells (e.g., triple-negative breast cancer) [21] (at their half maximal inhibitory concentration (IC_50_) values). FTIR and Raman microspectroscopies were applied to identify possible biomarkers of cytotoxicity, aiming at the development of new and improved chemotherapeutic drugs. The metabolic profile of each administered drug against each prostate cancer cell line (LNCaP and PC-3) was assessed, as well as the drugs´ impact on a healthy prostate cell line (PNT-2).

## 2. Results and Discussion

### 2.1. Cytotoxic Evaluation

The cytotoxic effect of cisplatin and Pd_2_Spm was evaluated through the MTT assay, against two prostate cancer cell lines (PC-3 and LNCaP) and one healthy prostate cell line (PNT-2) (Figure 2). A clear separation between dose–response curves for prostate cancer and non-cancer cells was observed for Pd_2_Spm at 24, 48, and 72 h incubation times. For cisplatin, a slight overlap between all the three cell lines was detected at 24 and 48 h. Table 1 comprises the IC_50_ values obtained for cisplatin and Pd_2_Spm for 24, 48, and 72 h incubation periods. For cisplatin, the values obtained were lower for all time points when compared to Pd_2_Spm. The healthy cells (PNT-2) were more sensitive to both Pt(II)/Pd(II) compounds when compared to the corresponding cancer cells (PC-3 and LNCaP). This is indicative of a non-discriminatory cytotoxic effect induced by these metal-based drugs. Values previously reported for cisplatin for PC-3, LNCaP, and PNT-2 were: PC-3–1.0 µM [41] (72 h, SRB method) 10.6 µM [42] (48 h, MTT method), LNCaP—3.7 µM [41] (48 h, SRB method), and PNT-2–8.5 µM [43] (96 h, MTT method), which reflects some variability, often due to the type of method as well as the experimental design. The higher effect of cisplatin in the hormone-insensitive PC-3 cells, when compared to the hormone-sensitive LNCaP, was also presently identified (Table 1).

### 2.2. Raman and FTIR Microspectroscopy Characterization

When probing the biochemical signatures of cell lines by Raman and FTIR microspectroscopies, a large number of data points are usually required. This is important in order to average the resulting spectral signatures from each individual cell and to obtain a representative spectral signature of the sample (Figure 3), thus minimizing the effect of the cell cycle profile (predominantly G_1_ under normal conditions) as previously reported [39]. In this study, both FTIR and Raman data were acquired through a randomly distributed cell population consisting of cells exhibiting a high spectral heterogeneity due to different stages of the cell cycle.

Figure 3 depicts the average Raman and FTIR spectra obtained for the three prostate cell lines (PNT-2, LNCaP, and PC-3) under the following conditions: (1) untreated cells (control), and (2) treated cells (either with cisplatin or Pd_2_Spm). In Appendix A, the assignments for all the signals observed in the FTIR and Raman spectra for control cells of PNT-2, LNCaP, and PC-3 cell lines are identified (including signals only detected by infrared, and specific DNA and protein conformational rearrangements promoted by drug exposure) [21,22]. The spectra for untreated cells are in accordance with published data based on FTIR microspectroscopy [38,40].

The cells were fixed with a 4% formaldehyde solution (formalin), considered the optimal method for cellular fixation in order to preserve the sample in a similar physiologic state condition, avoiding contaminations in the fixation process. Formalin leads to the formation of methylene bridges between aldehyde groups from formaldehyde and the primary and secondary amines of cellular proteins, which allows to maintain a similar constitution to that of in vivo cells. A slightly less intense signal from the infrared spectra is expected from this fixation, caused by the conformational change of proteins and lipidic breakdown.

Principal components analysis (PCA) of the FTIR and Raman spectroscopic results was performed in order to unveil the chemical differences between control and drug-treated cells for each cell line under study (Figure 4). Overall, a good discrimination was attained by FTIR between controls, Pd_2_Spm-treated, and cisplatin-treated cells for each cell line (Figure 4A,C,E), this separation not being so clear for the Raman data (Figure 4B,D,F). A slight overlap between the scores of the control and drug-treated classes was expected, evidencing that the drugs did not affect the cells substantially towards complete destruction [44]. From Figure 4A, it is clear that the main discrimination between control, Pd_2_Spm-treated, and cisplatin-treated LNCaP cells in the FTIR spectra was along principal components 2 (PC2) and 3 (PC3) (explaining 39.8% and 11.3% of total variance, respectively). From these scores and loadings, it is possible to observe that discrimination between the drug-treated vs. control LNCaP cells in FTIR is mainly attributed to: drug-treated LNCaP cells showing more contribution, than control LNCaP cells, from amide III β-sheets at 1228 cm^−1^; amide II ((δ(CN-H)/ν(CN)), lipids (ν(CC), ν(CN), δ(CH_2_)), and proteins (δ(CH_2_), ν(=C=C=)_conjugated_) at 1547 cm^−1^; carbohydrates (δ(CH_2_)) at 1157 cm^−1^ and 1315 cm^−1^; and guanine (ν(CC)_ring_) at 1315 cm^−1^. Drug-induced conformational changes were also detected: the bands attributed to amide I anti-parallel β-sheets (1691 cm^−1^) were stronger in the drug-treated cells, reflecting a clear effect on proteins. The LNCaP control group showed a higher intensity than drug-treated cells in the bands assigned to proteins (ν(CC), ν(CN)), phospholipids (ν_s_(PO_2_^−^), ν(CC), ν(CO)), and carbohydrates (δ(OCH), ν(CC), ν(CO)_glycogen_), detected at 1050 cm^−1^ and 1074 cm^−1^, which may be indicative of a drug interaction with the cellular membrane (phospholipid moieties).

Regarding the Raman spectra of the LNCaP cells, the classes were not as distinctly separated as in FTIR (Figure 4B): PC1 and PC2 providing a slight separation and explaining 31.4% and 24.8% of the total variance, respectively. As for FTIR, the LNCaP control cells showed a wider dispersion in both principal components relative to the drug-treated cells. Relative to the control, drug-treated LNCaP cells evidenced: a higher contribution from Z-DNA (ν(OPO)_backbone_) at 749 cm^−1^; proteins (ν(CN)) at 1128 cm^−1^; guanine (ν(CC)_ring_) at 1328 cm^−1^; cytosine, guanine, and thymine at 1347 cm^−1^; adenine, guanine, thymine, glycoproteins (δ(CH_3_)), and lipids/acyl chains (δ(CH_2_)) at 1379 cm^−1^; and amide I antiparallel β-sheets at 1680 cm^−1^. The drug-treated LNCaP cells have also suffered a shift of the phenylalanine band from 1001 cm^−1^ to 1008 cm^−1^. The DNA (OPO) backbone elongation is the most sensitive peak to recognize cell death since this indicates a collapse of the phosphodiester bonds in the double helix [21].

LNCaP control cells showed a higher contribution, compared to drug-treated cells, from B-DNA/deoxyribose (ν(CO)) and protein/lipids/carbohydrates (ν(CC), ν(CN), and ν(CC), ν(CO)) at 1064 cm^−1^; RNA/adenine and cytosine (ν(CC)_ring_) at 1297 cm^−1^; lipids (δ(CH_2_)) at 1440 cm^−1^; and amide I (random coil) at 1655 cm^−1^.

Regarding the PC-3 cell line, a separation between Pd_2_Spm-treated cells vs. the remaining groups (control and cisplatin-treated being overlapped) was clear in the FTIR spectra (Figure 4C), along a combination of PC1 and PC2 (representing 45.1% and 26.5% of total data variance, respectively). This separation is mainly due to a stronger contribution, in Pd_2_Spm-treated PC-3 cells (compared to PC-3 controls and PC-3 cisplatin-treated cells), from phenylalanine (δ(CH), ν(O–CH_3_)), phospholipids (ν(CC), δ(CH)), and carbohydrates (ν(CC), ν(CO),ν(C–OH)) at 1033 cm^−1^; porphyrins (ν(C=C)) at 1527 cm^−1^; amide II (δ(CN-H)/ν(CN)) at 1551 cm^−1^; and amide I (random coil) at 1659 cm^−1^.

From the Raman spectra, separation between Pd_2_Spm-treated PC-3 cells vs. PC-3 control and PC-3 cisplatin-treated cells was obtained along PC5 (5.0% of total variance, Figure 4D). As in FTIR, PC-3 controls and cisplatin-treated PC-3 cells were slightly overlapped. The Pd_2_Spm-treated PC-3 cells presented a higher contribution from cytosine and adenine (ν(CC)_ring_) at 779 cm^−1^; guanine at 1333 cm^−1^; adenine and guanine (ν(CC)_ring_) at 1571 cm^−1^; phenylalanine (ν_s_(CC)_ring_) at 1003 cm^−1^; lipids (δ(CH_2_)) at 1433 cm^−1^ and (ν(C=C)) at 1650 cm^−1^; and amide I (α-helix) at 1651 cm^−1^.

Finally, regarding the non-cancer prostate cell line, PNT-2, it is visible that the discrimination between the three groups (control, Pd_2_-treated and cisplatin-treated cells) in the FTIR spectra occurs along both PC2 and PC3, covering 7.1% and 4.9% of total data variance, respectively (Figure 4E). The drug-treated PNT-2 cells show a higher contribution from lipids (δ(CH_2_)) at 1453 cm^−1^; porphyrins (ν(C=C)) at 1526 cm^−1^; and random coil and β-sheets of amide I at 1650 cm^−1^. When compared to the drug-treated samples, the PNT-2 control cells evidence a higher contribution from proteins and membrane lipids (δ(CH_2_), ρ(CH_2_)) at 1402 cm^−1^; lipids (δ(CH_2_) at 1455 cm^−1^); porphyrins (ν(C=C) at 1522 cm^−1^) and nucleic acids (ν_s_(PO_2_^-^)_B-DNA_ at 1086 cm^−1^), adenine and cytosine (ν(CC)_ring_ at 1577 cm^−1^).

From PNT-2 Raman data (Figure 4F), it is possible to separate the control from the drug-treated cells (Pd_2_Spm and cisplatin-treated cells are overlapping) along PC4 (which covers 10.3% of the total variance). The drug-exposed PNT-2 cells show a higher contribution from thymine (ν(CC)_ring_) at 1602 cm^−1^, and proteins (ν(CS), τ(CC)_tyrosine_ at 644 cm^−1^, δ(C=CH)_phenylalanine_ at 1604 cm^−1^, and amide I (random coil) at 1648 cm^−1^).

From Figure 4A,E, it is visible that the FTIR scores from Pd_2_Spm-treated LNCaP and PNT-2 cells display an intermediate position relative to the cisplatin-treated and control samples, as opposed to the PC-3 cells. This may suggest that different metabolic pathways might be involved in the cytotoxic activity of both Pt-drugs drugs. The same intermediate position of Pd_2_Spm-treated cells is visible in the Raman scores of LNCaP (Figure 4B), although this effect is not evident for the other cell lines when probed by Raman spectroscopy.

For a deeper interpretation of the effect of cisplatin and Pd_2_Spm on prostate healthy vs. cancer cell lines (PNT-2 vs. PC-3 and LNCaP), the principal component analysis of the Raman and FTIR data is shown in Figure 5. For the Pd_2_Spm-treated PC-3 vs. PNT-2 (Figure 5A,B), the FTIR spectra were distinctly separated along PC3 (explaining 4.2% of the total variance). Based on the Raman data, these groups were separated along PC1 (which covered 48.0% of the data variance). Therefore, it was clear that the Pd_2_Spm-treated PC-3 cells presented a higher contribution from proteins (ν(O-CH_3_) and δ(CH)_phenylalanine_ at 1029 cm^−1^; ν(CC), ν(CN), δ(CH_2_) at 1152 cm^−1^) in FTIR; a blue-shift of ν(CC), ν(CO), ν(C–OH) from phenylalanine, phospholipids, and carbohydrates to 1008 cm^−1^ in Raman; lipids (δ(CH_2_), ν(=C=C=)_conjugated_) at 1152 cm^−1^, and δ(CH_2_) at 1323 cm^−1^, in FTIR, and at 1442 cm^−1^ (δ(CH_2_)) in Raman; amide I (antiparallel β-sheets at 1672 cm^−1^ in FTIR and random coil at 1659 cm^−1^ in Raman); ν(C=O) of amino acids’ side chain at 1691 cm^−1^ in FTIR; and guanine (ν(CC)_ring_) at 1323 cm^−1^ in FTIR.

Regarding the Pd_2_Spm-treated PNT-2 vs. PC-3 cells, the former presented a higher contribution, in the FTIR spectra, from B-DNA ν(CO)_deoxyribose_ at 1058 cm^−1^ and ν_s_(PO_2_^−^) at 1085 cm^−1^, amide II (δ(CN-H), ν(CN)) at 1542 cm^−1^, amide I (parallel β-sheet) at 1639 cm^−1^, and amide I (random coil) at 1649 cm^−1^. From the Raman spectra, PNT-2 presented higher intensity from B-DNA/nucleic bases and tryptophan (ν(CC)_ring_) at 678 cm^−1^, B-DNA/deoxythimine and tryptophan (ν(CC)_ring_) and Z-DNA (ν(OPO)_backbone_) at 753 cm^−1^, nucleic acids (ν(CC)_ring_ from cytosine, thymine, and uracil) at 784 cm^−1^, ν(CC), ν(CO), ν(C-OH) from phenylalanine, phospholipids, and carbohydrates at 1000 cm^−1^, proteins and lipids (ν(C=C), ν(C=N)) at 1583 cm^−1^.

When comparing cisplatin-treated PC-3 cells vs. cisplatin-treated PNT-2 cells (Figure 5C,D), separation was obtained along PC2 in FTIR (32.4% of total variance), and along PC1 and PC3 in Raman (corresponding to 29.5% and 18.8%, respectively). The major biomarkers for separation were: PNT-2 showed a higher intensity in the bands assigned to phenylalanine (δ(CH)) at 1024 cm^−1^ in FTIR and at 1001 cm^−1^ and 1603 cm^−1^ in Raman; B-DNA/deoxythimine at 677 cm^−1^ and 752 cm^−1^; tryptophan (ν(CC)_ring_) and porphyrin (ν(C=C)) at 1556 cm^−1^ in Raman. PC-3 cell line presented a higher contribution from A-DNA (δ(CH_2_) and ν(C=O)) at 1414 cm^−1^ and 1708 cm^−1^ in FTIR, and in Raman, showed the band assigned to phospholipids and phenylalanine blue-shifted to 1008 cm^−1^, and a higher contribution from RNA/ribose (ν(CO)), proteins (ν(CN)), and lipids (ν(CC)_acyl (*trans* conformation)_) at 1132 cm^−1^, amide III/α-helix and lipids (ω(CH_2_), t(CH_2_)) at 1277 cm^−1^, RNA/adenine and cytosine at 1300 cm^−1^, and amide I random coil at 1659 cm^−1^.

It is noteworthy that even though the FTIR loadings of the principal components that separate the PNT-2 vs. PC-3 cells treated with either Pd_2_Spm or cisplatin (Figure 5A,C, respectively) display a similar information, with the bands ascribed to aromatic lipids and proteins (at ca. 1466 cm^−1^) being more clearly observed in the cisplatin-treated PC-3 cells than in the Pd_2_Spm-treated ones. Moreover, it is noticeable that the Pd_2_Spm-treated healthy prostate cell line PNT-2 displays a strong contribution from B-DNA deoxyribose (ca. 1059 cm^−1^) and B-DNA phosphates (ca. 1087 cm^−1^) (Figure 5A), which is not verified for the prostate cancer cell line PC-3. When these cells were treated with cisplatin, PNT-2 showed a lower contribution from B-DNA deoxyribose and B-DNA phosphates than PC-3 (Figure 5C). Using Raman spectroscopy, the first principal component (PC1) in the Pd_2_Spm-treated cells was relevant for PC-3 vs. PNT-2 separation (Figure 5B) and its loading is comparable to the PC1 loading of the cisplatin-treated cells (Figure 5D). However, for the latter, the third principal component was more determinant for discrimination of these two cell lines than PC1 alone, giving some insight into the effect of cisplatin—namely PNT-2 cells displaying a stronger intensity of B-DNA/deoxythimine (ca. 753 cm^−1^), RNA/ribose (ca. 1131 cm^−1^) than PC-3 cells.

Regarding the LNCaP vs. PNT-2 Pd_2_Spm-treated cells, a clear separation was obtained with both FTIR and Raman, along PC1 (75.3% and 43.4% of the total variance, respectively) (Figure 5E,F). The main contributions for this separation were: a higher intensity of the bands assigned to B-DNA/deoxyribose (at 1059 cm^−1^ and 1086 cm^−1^ in FTIR), amide I (β-sheets) at 1638 cm^−1^ in FTIR and at 1675 cm^−1^ in Raman, amide II at 1545 cm^−1^ in FTIR and aromatic lipids (at 1465 cm^−1^ in FTIR), for Pd_2_Spm-treated LNCaP cells as compared to Pd_2_Spm-treated PNT-2.

Similarly to cisplatin-treated LNCaP vs. PNT-2 cells (Figure 5G,H), a good separation was obtained along PC1 in FTIR (covering 78.4% of total variance) and along PC2 in Raman (corresponding to 26.6% of total variance). Relative to cisplatin, the PNT-2 cells showed a higher intensity from B-DNA (ν_s_(PO_2_^-^)) at 1087 cm^−1^ in FTIR and at 1095 cm^−1^ in Raman; phenylalanine (δ(CH)) at 1031 cm^−1^ in FTIR and at 1171 cm^−1^ in Raman; phospholipids (δ(CH)) and carbohydrates (ν(CC), ν(CO), ν(COH)) at 1031 cm^−1^ in FTIR and at 1001 cm^−1^ in Raman; lipids (δ(CH_2_)) at 1439 cm^−1^; and amide I (random coil) at 1657 cm^−1^ in Raman. In contrast, the Raman signal from cisplatin-treated LNCaP cells showed a stronger contribution, relative to PNT-2, from RNA/ribose (ν(CO)), lipids (ν(CC)_acyl-*trans*_), and proteins (ν(CN)) at 1133 cm^−1^, tyrosine proteins (ν(C=C), ν(C=N)) and lipids (ν(C=C), ν(C=N)) at 1588 cm^−1^.

It is noteworthy that for LNCaP vs. PNT-2, the principal components that separate both cell lines treated with either Pd_2_Spm (PC1 in Figure 5E) or cisplatin (PC1 in Figure 5G) are very similar in FTIR. The main difference is the band assigned to amide II at 1545 cm^−1^, which plays a more important role in cisplatin-treated than in Pd_2_Spm-treated LNCaP cells.

## 3. Materials and Methods

### 3.1. Chemicals and Solutions

Cisplatin (cis-dichlorodiammine platinum(II), >99.9%), (3-(4,5-dimethylthiazol-2-yl)-2,5-diphenyltetrazolium bromide (MTT), dimethyl-sulphoxide (DMSO), penicillin/streptomycin 100× solution, phosphate buffered saline (PBS), potassium tetrachloropalladate(II) (K_2_PdCl_4_, >99.9%), RPMI-1640 medium, N,N´-bis(3-aminopropyl)-1,4-diaminobutane (spermine, free base, Spm), Trypan blue (0.4% *w*/*v*), trypsin-EDTA (1×), as well as solvents, inorganic salts and acids, were purchased from Sigma-Aldrich Chemical S.A. (Sintra, Portugal). Fetal bovine serum (FBS) was obtained from Gibco-Life Technologies (Porto, Portugal). All reagents were of analytical grade. Pd_2_Spm was synthesized according to published procedures [45] optimized by the authors [46].

For drug administration, initial stock solutions of cisplatin (320 µM) were prepared in PBS and Pd_2_Spm (320 µM) was prepared in PBS with a 10% (*v*/*v*) DMSO. All solutions were sterile filtered and stored at −20 °C.

### 3.2. Cell Culture

The human prostate cancer (PC-3 and LNCaP) and non-tumorigenic (PNT-2) cell lines were obtained from the ECCAC Culture Collections and supplied by Sigma-Aldrich Chemical S.A. (Sintra, Portugal). They were cultured as monolayers, at 37 °C, in a humidified atmosphere of 5% CO_2_. The cultures were maintained in RPMI-1640 culture medium, supplemented with 10% (*v*/*v*) heat-inactivated FBS and 1% (*v*/*v*) penicillin/streptomycin. The cells were subcultured at 80% confluence, using 0.05% trypsin-EDTA (1×) in PBS.

### 3.3. Cytotoxic Evaluation

For all cell lines (PC-3, LNCaP and PNT-2), cell cultures were established in 96-well plates (100 µL/well) at a density of 3 × 10^4^ cells/cm^2^, and were allowed to attach for 24 h. Triplicates were treated for different incubation periods (three independent experiments) with a range of concentrations between 2 and 32 µM of the tested compounds (cisplatin and Pd_2_Spm). According to the population doubling time for all cell lines (ca. 24 h), 24, 48, and 72 h timepoints after drug administration were chosen.

In order to evaluate cell viability, the MTT assay was used. Briefly, at each time-point, the growth media was removed, cells were washed with PBS and MTT solution (2.5 mg/mL) was added to each well (100 µL) [47]. After 3 h of incubation at 37 °C, the formazan crystals were solubilized in DMSO (100 µL), and the absorbance was measured at 570 nm.

### 3.4. Sample Preparation for Vibrational Microspectroscopies

Upon harvesting by trypsinization, all prostate cell lines (PC-3, LNCaP, and PNT-2) were centrifuged and the pellet was resuspended in culture medium and seeded, at a concentration of 3 × 10^4^ cells/cm^2^, on optical substrates suitable for either FTIR or Raman data collection, respectively, CaF_2_ disks (Crystran UV-grade, 1 mm × 13 mm) or MgF_2_ disks (Crystran Raman-grade, 1 mm × 13 mm), which were previously washed with 70% ethanol. After incubation for 24 h (allowing the cells to adhere), each of the tested drugs (cisplatin or Pd_2_Spm) were added according to the respective 50% cell growth inhibition values (IC_50_) for each cell line, and the cells were allowed to culture for a further 48 h. The growth medium was then removed, the cells were washed twice with 0.9% NaCl (*w*/*v*), fixed in 4% formalin (diluted in 0.9% NaCl from the commercial neutral buffered formaldehyde solution) for 10 min, and washed several times with deionized water (to remove any residual salt) [48]. The disks were allowed to air-dry prior to spectroscopic analysis.

All samples were prepared in duplicate, in two independent experiments.

### 3.5. FTIR Microspectroscopy

FTIR acquisitions were performed using a Bruker Hyperion 2000 microscope with a liquid nitrogen-cooled mercury-cadmium-telluride (MCT) detector, in transmission mode using a 15× Cassegrain both condenser and objective, coupled to a Bruker Optics Vertex 70 spectrometer, both purged by CO_2_-free dry air. Each spectrum was the sum of 128 scans, at 4 cm^−1^ resolution, and the 3-term Blackman–Harris apodization function was applied.

### 3.6. Raman Microspectroscopy

Raman microspectroscopy acquisition was performed using a WITec confocal Raman microscopy system Alpha 300 R coupled to an ultra-high-throughput spectrometer (UHTS) 300 VIS-NIR, using a 532 nm diode-pumped solid-state laser. The laser power on the sample was kept at 28 mW and the measurements were achieved using a 100×/0.8 Zeiss Epiplan objective with 10 accumulations of 10 s per spectrum for PC-3 cell line, and 15 accumulations of 10 s per spectrum for LNCaP and PNT-2 cell lines.

### 3.7. Data Processing and Statistical Analysis

Cells viability data results are expressed as mean ± standard error of the mean (SEM) and compared with non-treated controls. Statistical analysis was carried out through one-way ANOVA followed by Tukey’s multiple comparison test. A *p*-value < 0.05 was considered statistically significant.

The microFTIR spectral data, processed by the OPUS 9.1 software, was atmospherically compensated and corrected for changes in background intensity. The quality of the spectra was assessed based on the intensity of the amide I band. The spectra were corrected for Mie scattering using an Extended Multiplicative Signal Correction (EMSC) function with 20 iterations [49]. The analyzed spectra were truncated from 1000 to 1800 cm^−1^ and vector normalized. These operations were performed in Matlab2020b (MathWorks).

The microRaman spectral data were obtained using the Project FIVE software (WITec). In order to reduce the spectral noise, principal component noise removal was performed, retaining the first 20 principal components. The spectral data were cropped to the fingerprint region, 500–1800 cm^−1^, and further corrected for cosmic ray removal, background subtraction (polynomial order fitting of 2nd order), Savitzky–Golay smooth filtered (window width of 5 points), and the area was normalized. The data pre-processing was performed using the Quasar Spectroscopy 1.6.0 software [50,51]. In order to analyze the spectral information between cell line classes and to identify possible biomarkers, principal component analysis was done. The order of the principal components (PC) denoted their importance in relation to the data set variance, PC1 corresponding to the highest variance present in the data.

## 4. Conclusions

The development of new and improved chemotherapeutic drugs is aimed at achieving both efficacy and selectivity, i.e., leading to irreversible damage to cancer cells while maintaining the integrity of healthy cells. In addition, acquired resistance to treatment should be minimized, as well as deleterious side effects, by exploring novel pathways of cytotoxicity. The impact of Pd_2_Spm towards human prostate cancer cell lines (LNCaP and PC-3) and human prostate non-cancer cells (PNT-2) was studied (and compared to cisplatin´s effect), through the evaluation of the cellular viability (by biological assays) coupled to spectroscopic measurements of the cells’ biochemical profile (by FTIR and Raman microspectroscopies).

A non-discriminatory cytotoxic effect induced by both Pt- and Pd-agents was observed in all tested cell lines. Still, considering the least possible damage to non-cancer cells (PNT-2) prompted by drug exposure, cisplatin appeared to be the best choice when compared to Pd_2_Spm after 48 h of drug administration (IC_50_ = 5.2 µM vs. 9.0 µM). Concerning prostate cancer, an IC_50_ of 3.0 µM was obtained for the PC-3 cell line, which is lower than the 5.2 µM value found for the PNT-2 non-cancerous cells.

Regarding the spectroscopic data, drug-treated cells evidenced key changes such as: native B-DNA to Z- or A-DNA conformations; shifts to high wavenumbers of bands from phenylalanine (ν_s_(CC)_ring_) and phospholipids (δ(CH)). Additionally, the ν(C=O) signal from ester groups suggested a clear perturbation of the cellular membrane, particularly in the nonpolar part of the phospholipids, which may affect the membrane integrity.

The different cytotoxic activity induced by cisplatin vs. Pd_2_Spm is suggested by the intermediate position that Pd_2_Spm-treated LNCaP and PNT-2 cells present relative to the corresponding cisplatin-treated and control samples. A higher amount of lipids was detected in the presence of cisplatin as compared to the Pd-agent, suggesting a drug interaction with the cellular membrane. Additionally, Pd_2_Spm showed a more significant impact on proteins and carbohydrates. Regarding the Pd_2_Spm-treated non-cancer PNT-2 cells, a strong contribution from B-DNA deoxyribose and B-DNA phosphates was found. However, when PNT-2 and PC-3 cell lines were treated with cisplatin, the former evidenced a lower contribution from B-DNA deoxyribose and B-DNA phosphates, while the PC-3 cells show a conformational change from native B-DNA to A-DNA (indicative of DNA disruption).

MicroFTIR and microRaman are two powerful analytical techniques which enabled us to conclude that the two drugs currently tested behave differently against the human prostate cell lines under study. These findings on the metabolic impact of anticancer Pt- and Pd-complexes are expected to contribute to the development of new and optimized metal-based agents for specific cancer types as well as subtypes.

## Figures and Tables

**Figure 1 ijms-24-01888-f001:**
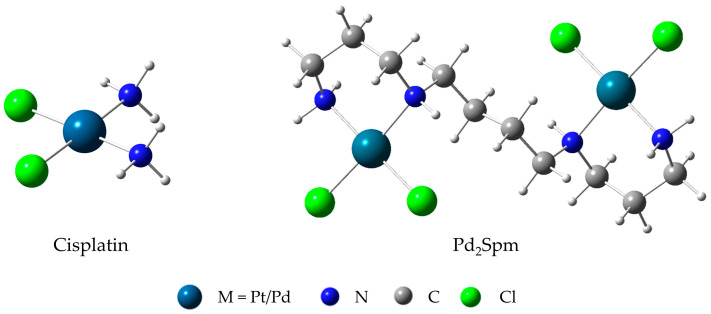
Structural representation of the Pt(II) and Pd(II) complexes presently studied.

**Figure 2 ijms-24-01888-f002:**
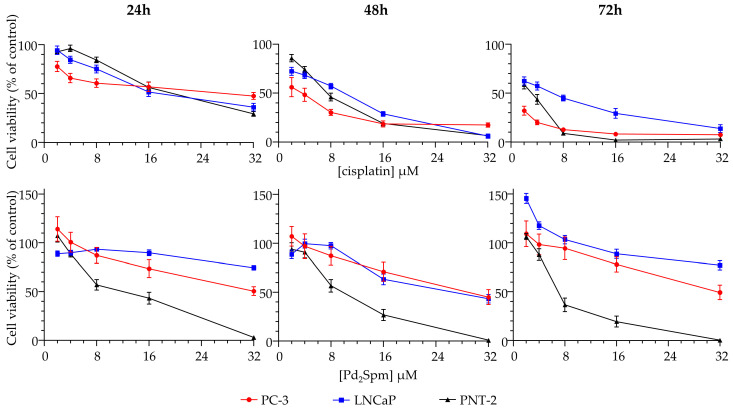
Dose–response curves of cisplatin and Pd_2_Spm against prostate cancer (PC-3, red line; LNCaP, blue line) and prostate non-cancer (PNT-2, black line) cell lines, at 24, 48, and 72 h incubation times. Data are expressed as mean ± SEM, *n* = 4.

**Figure 3 ijms-24-01888-f003:**
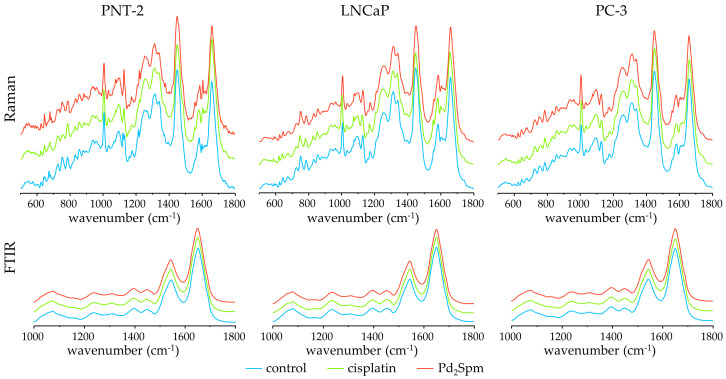
Mean Raman (500–1800 cm^−1^) and infrared (1000–1800 cm^−1^) spectra for untreated/control (blue line), cisplatin-treated (green line), and Pd_2_Spm-treated (red line) cells.

**Figure 4 ijms-24-01888-f004:**
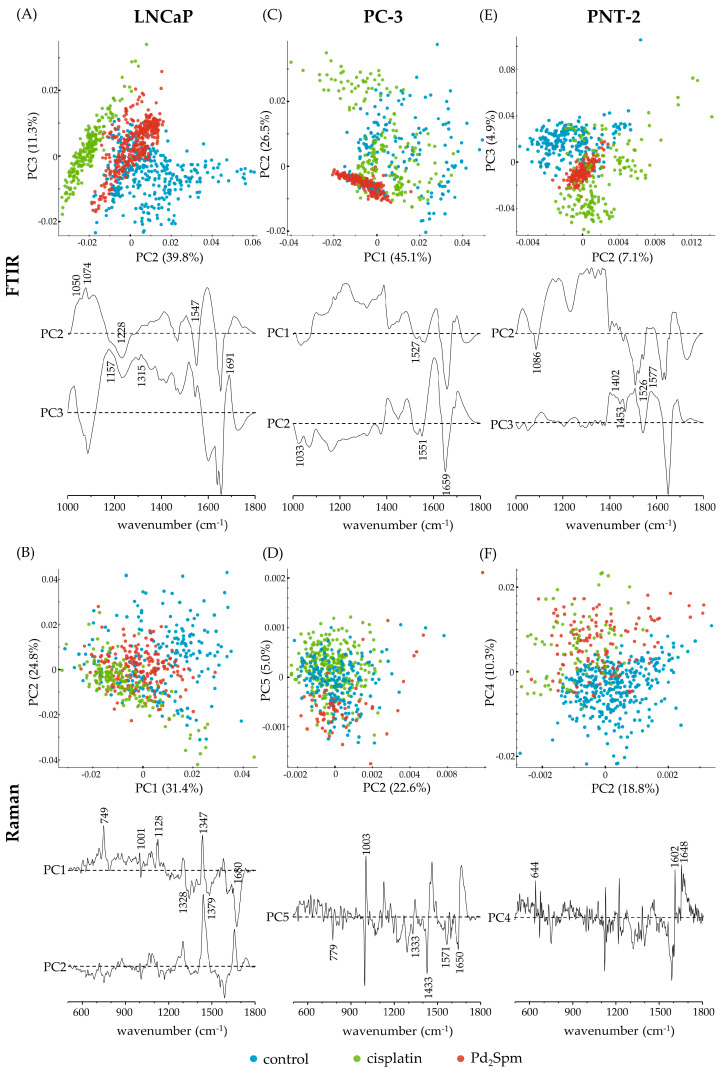
PCA scores and loading plots of FTIR ((**A**,**C**,**E**); 1000–1800 cm^−1^) and Raman ((**B**,**D**,**F**); 500–1800 cm^−1^) data for cisplatin and Pd_2_Spm-treated prostate cancer cell lines LNCaP (**A**,**B**) and PC-3 (**C**,**D**), and non-cancer cell line PNT-2 (**E**,**F**) vs. their respective controls. (For clarity the loadings are offset, the dashed horizontal lines indicating zero loading).

**Figure 5 ijms-24-01888-f005:**
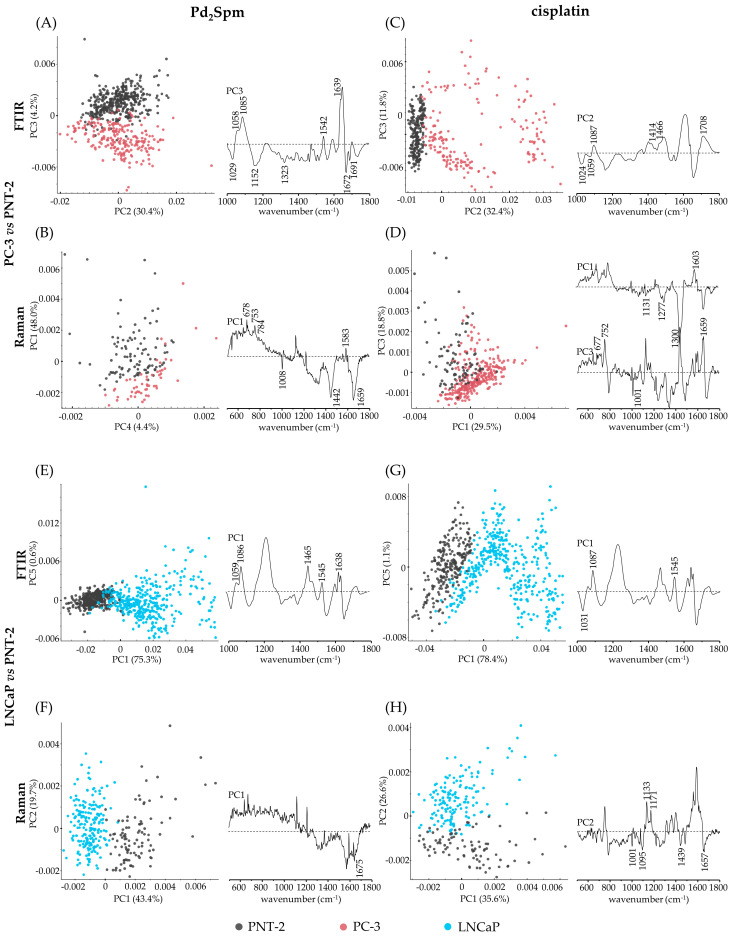
PCA scores and loading plots of FTIR ((**A**,**C**,**E**,**G**); 1000–1800 cm^−1^) and Raman ((**B**,**D**,**F**,**H**); 500–1800 cm^−1^) data for cisplatin- and Pd_2_Spm-treated prostate cancer cell lines PC-3 (**A**–**D**) and LNCaP (**E**,**F**,**G**,**H**) vs. the non-cancer cell line PNT-2. (For clarity, the loadings are offset, the dashed horizontal lines indicating zero loading).

**Table 1 ijms-24-01888-t001:** Half maximal inhibitory concentration (IC_50_, μM) of cisplatin and Pd_2_Spm against prostate cancer (PC-3 and LNCaP) and prostate non-cancer (PNT-2) cell lines, at 24, 48, and 72 h incubation times.

Drug	Time	PC-3	LNCaP	PNT-2
Cisplatin	24 h	25.6	13.6	13.2
	48 h	3.0	7.3	5.2
	72 h	0.7	2.7	2.4
Pd_2_Spm	24 h	31.3	76.0	6.6
	48 h	27.9	27.0	9.0
	72 h	31.2	28.5	5.4

## Data Availability

The data presented in this study are available on request from the corresponding author.

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
