# Peer review of "Evaluation of the Cytotoxic Effect of Pd2Spm against Prostate Cancer through Vibrational Microspectroscopies"

_ijms, 2023, doi:10.3390/ijms24031888_

Round 1

Reviewer 1 Report

Reviewer #1: Critical review on manuscript entitled:

Evaluation of the Cytotoxic Effect of Pd2Spm Against Prostate Cancer Through Vibrational Microspectroscopies

In this article, authors study, a complex with two Pd(II) centres linked by the biogenic polyamine spermine (Pd2Spm), was tested on healthy (PNT-2) and cancer (LNCaP and PC-3) prostate human cell lines, using cisplatin as a reference by using FTIR and Raman microscpectroscopies.

The article is fairly well written, and the topic could be of great interest to the scientific community. However, I believe that, at the present stage, the work still presents critical structural and methodological flaws. Therefore, I would not recommend its publication in its current form.

Here below, the main criticalities are summarized, hoping they may be useful to improve the manuscript in case of a future resubmission.

1.     Introduction section: The introduction section should be expanded. Authors should describe the current knowledge on the use of Raman spectroscopy and FTIR in prostate cancer studies.

2.     Materials and methods section: Please describe which testing model was used in chemometric methods.

3.     Fig. 3 Please add the scale on the y-axis and mark the peaks.

4.     Fig. 4 Please mark the peaks on loading plots of FTIR and Raman. The same for Fig. 5

5.     The literature review is not sufficient.

6.     Please compare your results with the literature.

7.     Please provide a sensitivity, specificity and confusion matrix for obtained chemometric analysis.

Author Response

  1. Introduction section: The introduction section should be expanded. Authors should describe the current knowledge on the use of Raman spectroscopy and FTIR in prostate cancer studies.

Response 1: The authors appreciate your comments and suggestions. The authors have added more information on the Pt-based compounds under clinical studies and have also completed the section regarding the use of Raman and FTIR in prostate cancer.

  1. Materials and methods section: Please describe which testing model was used in chemometric methods.

Response 2:  We believe there is a misinterpretation of our data analysis procedure. In this manuscript the only chemometric method used was Principal Component Analysis (PCA) to interpret the spectral information that differ between the groups, as stated in the methods section (subsection 3.7 Data Processing and Statistical Analysis): “In order to analyse the spectral information between cell line classes and to identify possible biomarkers, principal component analysis was done. The order of the principal components (PC) denoted their importance in relation to the data set variance, PC1 corresponding to the highest variance present in the data.”. We did not perform any classification model development (with model training and respective testing and validation).

  1. Fig. 3 Please add the scale on the y-axis and mark the peaks.

Response 3: The intensity units used in Raman and FTIR spectroscopies are relative arbitrary units, therefore they do not hold a significance on its own, especially after data pre-processing.

  1. Fig. 4 Please mark the peaks on loading plots of FTIR and Raman. The same for Fig. 5

Response 4: The authors appreciate this suggestion and have included the wavenumbers in the figures as requested.

  1. The literature review is not sufficient.

Response 5: The literature review has been extended and more references were added accordingly (currently a total of 51 references).

  1. Please compare your results with the literature.

Response 6: Subsection 2.1 was reviewed and the results extensively compared to what is presently found in the literature for cisplatin against prostate cancer. Regarding the vibrational spectroscopy results, there are not any comparable published studies, to the best of the authors’ knowledge, to establish a direct comparison (same cell lines, similar compounds and data analysis procedure).

  1. Please provide a sensitivity, specificity and confusion matrix for obtained chemometric analysis.

Response 7: As explained in Q2, we have not developed a classification model and respective testing. Therefore, we do not have any confusion matrix, nor sensitivity/specificity, as these are metrics of the performance of a classification model.  

Reviewer 2 Report

In this paper, the cytotoxic effects of Pd2Spm and cisplatin on prostate cancer were studied using FTIR microspectroscopy and Raman microspectroscopy combined with PCA model. With some modifications, it can be published in ‘International Journal of Molecular Sciences’. Both drugs rely on DNA, lipids and proteins as biomarkers. The main change is between the B-DNA protoconformation and Z-DNA and A-DNA, where cisplatin has a greater effect on lipids than Pd2Spm.

1、              In the description of Table 1, for cisplatin, compared with Pd2Spm, the values obtained at all time points are lower, but Table 1 is inconsistent with this description (Lines 7-8 of Part 2.1).

2、              All abbreviations that appear in the manuscript should be explained when they are first presented.

3、              FT-IR in Fig. 3: please flip the x-axis. The wavenumber range shall start with the highest wavenumbers to the lowest wavenumbers.

4、              The measurement method of FTIR microspectroscopy is not clearly introduced in the paper. Is the measurement method transmission or reflection? How to use CaF2 disks to measure?

Author Response

1. In the description of Table 1, for cisplatin, compared with Pd2Spm, the values obtained at all time points are lower, but Table 1 is inconsistent with this description (Lines 7-8 of Part 2.1).

Response 1: The authors would like to thank the reviewer for the positive comments and suggestions, which will improve the quality of the manuscript. The text was rephrased for better understanding.

2. All abbreviations that appear in the manuscript should be explained when they are first presented.

Response 2: All abbreviations were carefully checked and are now introduced properly throughout the manuscript.

3、FT-IR in Fig. 3: please flip the x-axis. The wavenumber range shall start with the highest wavenumbers to the lowest wavenumbers.

Response 3: The authors appreciate this suggestion. However, as in this manuscript we also present Raman spectra we would prefer to have the x-axis for both Raman and FTIR plots in the same wavenumber ascending order, for clarity and consistency.

4、The measurement method of FTIR microspectroscopy is not clearly introduced in the paper. Is the measurement method transmission or reflection? How to use CaF2 disks to measure?

Response 4: In line 2 of subsection 3.5 (FTIR Microspectroscopy) it is stated that the FTIR measurements are in transmission mode. Concerning the disks, in subsection 3.4 we have thoroughly explained that the cells were seeded on CaF2 disks, which were previously washed with 70% ethanol. After incubation for 24 h (allowing the cells to adhere), each of the tested drugs (either cisplatin or Pd2Spm) was added according to their respective IC50, and the cells were allowed to culture for a further 48 h. The growth medium was then removed, the cells were washed twice, fixed in 4% formalin and washed several times with deionised water. The disks were allowed to air-dry prior to spectroscopic analysis. At the time of the FTIR measurements each CaF2 disk (with the fixed cells) was placed on the FTIR microscope stage.